


# Surface fluxes of bromoform and dibromomethane over the tropical western Pacific inferred from airborne in situ measurements

Liang Feng[1,2], Paul I. Palmer[1,2], Robyn Butler[2], Stephen J. Andrews[3], Elliot L. Atlas[4], Lucy J. Carpenter[3], Valeria Donets[4], Neil R. P. Harris[5], Ross J. Salawitch[6], Laura L. Pan[7], Sue M. Schauffler[7]

1) National Centre for Earth Observation, University of Edinburgh, Edinburgh, UK

2) School of GeoSciences, University of Edinburgh, Edinburgh, UK

3) Department of Chemistry, Wolfson Atmospheric Chemistry Laboratories, University of York, UK

4) University of Miami, Florida, USA

5) Centre for Atmospheric Informatics and Emissions Technology, Cranfield University, Cranfield, UK

6) University of Maryland, College Park, Maryland, USA

7) National Center for Atmospheric Research, Boulder, Colorado, USA

Corresponding author: Paul I. Palmer (paul.palmer@ed.ac.uk)

**ABSTRACT**

We infer surface fluxes of bromoform ($CHBr_3$) and dibromoform ($CH_2Br_2$) from aircraft observations over the western Pacific using a tagged version of the GEOS-Chem global 3-D atmospheric chemistry model and a Maximum A Posteriori inverse model. The distribution of *a priori* ocean emissions of these gases are reasonably consistent with observed atmospheric mole fractions of $CHBr_3$ (r=0.62) and $CH_2Br_2$ (r=0.38). These *a priori* emissions result in a positive model bias in $CHBr_3$ peaking in the marine boundary layer, but capture observed values of $CH_2Br_2$ with no significant bias by virtue of its longer atmospheric lifetime. Using GEOS-Chem, we find that observed variations in atmospheric $CHBr_3$ are determined equally by sources over the western Pacific and those outside the study region, but observed variations in $CH_2Br_2$ are determined mainly by sources outside the western Pacific. Numerical closed-loop experiments show that the spatial and temporal distribution of boundary layer aircraft data have the potential to substantially improve current knowledge of these fluxes, with improvements related to data density. Using the aircraft data, we estimate aggregated regional fluxes of $3.6\pm0.3\times10^8$ g/month and $0.7\pm0.1\times10^8$ g/month for $CHBr_3$ and $CH_2Br_2$ over $130^o$—$155^oE$ and $0^o$—$12^oN$, respectively, which represent reductions of 20—40% and substantial spatial deviations from the *a priori* inventory. We find no evidence to support a robust linear relationship between $CHBr_3$ and $CH_2Br_2$ oceanic emissions, as used by previous studies.



## 1.Introduction


The role of halogens in the catalytic destruction of stratospheric ozone is well established (WMO, 2014). The anthropogenic contribution to the inorganic halogen budget continues to decline in the stratosphere as a result of the Montreal protocol. A consequence of this

decline is that very short-lived substances (VSLS), halogenated compounds with e-folding lifetimes typically much less than 6 months, now represent a proportionally greater source of stratospheric halogens. The wide range of VSLS atmospheric lifetimes allow at least some of the emitted material to reach the upper troposphere, particularly over geographical regions where there is rapid, deep convection (Penkett et al., 1998; Yang et al., 2005; Warwick et al.,

2006; Levine et al., 2007; Pisso et al., 2010; Hosking et al., 2010; Carpenter et al., 2014; Hossaini et al., 2016a; Butler et al., 2016). Here, we use aircraft observations of bromoform ($CHBr_3$) and dibromomethane ($CH_2Br_2$) collected over the western Pacific Ocean to infer, using an inverse model, the magnitude and distribution of ocean emissions of these gases.

There are a wide range of VSLS that are beginning to limit the recovery of stratospheric ozone (e.g., Read et al., 2008; Hossaini et al., 2015; Oman et al., 2016). Chlorine VSLS are typically dominated by anthropogenic sources, but the fraction depends on the species (Hossaini et al., 2016b). Their natural sources include biomass burning, phytoplankton production, and soils. Iodine and bromine VSLS have predominately natural sources. Iodine

VSLS are mainly from ocean production processes, but with lifetimes of only a few days they are too reactive to be transported out of the marine boundary layer in large quantities. Bromine VSLS are also mainly from natural ocean sources (Gschwend et al., 1985; Manley et al., 1992; Sturges et al., 1992; Tokarczyk et al., 1994; Warwick et al., 2006; Carpenter and Liss, 2000; 2009; Palmer et al., 2009; Quack and Suess, 1999; Quack and Wallace, 2003;

Quack et al., 2007; Butler et al., 2007; Leedham et al., 2013). The most abundant bromine VSLS species are $CHBr_3$ and $CH_2Br_2$. Together they account for about 80% of bromine VSLS in the marine boundary layer (Law and Sturges, 2007; O'Brien et al., 2009; Hossaini et al., 2013). The local atmospheric lifetime for $CHBr_3$, determined by OH oxidation (76 days) and photolysis (36 days), is 24 days. $CH_2Br_2$ has a longer atmospheric lifetime of about 123

days, determined primarily by OH oxidation (123 days) and to a much lesser extent by photolysis (5000 days). Their lifetimes are sufficiently long that these natural halogenated compounds can be transported to the upper troposphere.



Previous measurement campaigns have reported that bromine VSLS and their degradation

products represent 2-8 pptv of stratospheric inorganic bromine (e.g., Dorf et al., 2008; Salawich et al., 2010). Complementary model simulations of atmospheric chemistry and transport, driven by *a priori* ocean emission inventories, report similar values (2-7 pptv) that are determined mainly by localized regions of active ocean biology that coincide with strong convection. Example regions include western Pacific Ocean, tropical Indian Ocean, and off

the Pacific coast of Mexico. These model calculations also suggest that 15-75% of the stratospheric bromine budget from bromine VSLS is delivered by the direct transport of the emitted halogenated compounds (Liang et al., 2010; Hossaini et al., 2016a; Aschmann et al., 2009). The large range of values reflects uncertainty in ocean emissions, model transport, and the wet deposition of degradation products in the upper troposphere lower stratosphere.


Current knowledge of ocean emissions of $CHBr_3$ and $CH_2Br_2$ are poorly constrained by the sparse measurements. Bottom-up and top-down methods have been used to estimate global $CHBr_3$ and $CH_2Br_2$ emissions. The bottom-up approach assumes local flux estimates are representation of larger spatial scales. Ship-borne air-sea flux observations with limited

spatial and temporal coverage are extrapolated over ocean basins (e.g. Quack and Wallace, 2003; Carpenter and Liss, 2000; Butler et al., 2007; Ziska et al., 2013). Poor observation coverage results in fluxes that rely heavily on assumptions used for extrapolation (Stemmler et al, 2015).

The top-down method, in this application, uses an atmospheric chemistry transport model to describe the relationship between emissions and the atmospheric measurements. The model emissions are fitted to the observations by adjusting their magnitude until the discrepancy between the model and observed atmospheric measurements is minimized. This fitting can be achieved using heuristic techniques or more established Bayesian

optimization methods (e.g. Liang et al, 2010, Ordonez et al, 2012, Ashfold et al., 2014). The short atmospheric lifetime of $CHBr_3$ poses particular difficulties for the top-down approach because atmospheric mole fractions are highly variable (Ashfold et al, 2014). Some studies have introduced (explicitly or implicitly) a simple linear correlation between $CHBr_3$ and $CH_2Br_2$ emissions to provide an additional constraint on the $CHBr_3$ flux estimate (e.g. Liang

et al., 2010, Ordóñez et al., 2012).  This approach, however, is then subject to errors associated with the assumption about the correlation. As with the bottom-up method, the top-down method is subject to errors due to poor spatial and temporal coverage of observations. By virtue of various assumptions made (and justified) by individual studies the resulting bottom-up and top-down $CHBr_3$ and $CH_2Br_2$ fluxes are significantly different (e.g.





Hossaini et al., 2016a). For example, the estimated global CHBr$_3$ annual emissions range from 216 Tg (Ziska et al., 2013) to 530 Tg (Ordóñez et al., 2012).

We use data from two coordinated aircraft campaigns over the western Pacific during 2014 to infer regional emission estimates of CHBr$_3$ and CH$_2$Br$_2$ for the campaign period using a
Bayesian inverse model. The Co-ordinated Airborne Studies in the Tropics (CAST; Harris et al., 2017), and CONvective Transport of Active Species in the Tropics (CONTRAST; Pan et al., 2016) campaigns measured a suite of trace gases and aerosols centred on the Micronesian region in the western Pacific, including Guam, Chuuk, and Palau during January and February 2014. We interpret aircraft measurements of CHBr$_3$ and CH$_2$Br$_2$ mole
fraction using the GEOS-Chem atmospheric chemistry transport model and a Maximum A Posteriori (MAP) inverse model approach.

In the next section we describe the CAST and CONTRAST CHBr$_3$ and CH$_2$Br$_2$ mole fraction data, the GEOS-Chem atmospheric chemistry transport model used to interpret the data,
and the MAP inverse model. In section 3, we report a model comparison with the CAST and CONTRAST atmospheric data, and results from the MAP inversion. We conclude the paper in section 4.

## 2. Data and Methods


We use CHBr$_3$ and CH$_2$Br$_2$ mole fraction determined using GC/MS from whole air sample (WAS) canisters collected during the CAST and CONTRAST aircraft campaigns during January 18[th] to February 28[th], 2014 (Harris et al., 2017; Pan et al., 2016). We refer the reader to Andrews et al. (2016) for a more detailed description of the observation data sets,
and to Butler et al (2016) for a statistical analysis of the CHBr$_3$ and CH$_2$Br$_2$ mole fraction data. For CAST, WAS canisters were filled aboard the Facility for Airborne Atmospheric Measurements (FAAM) BAe-146 UK Atmospheric Research Aircraft. These canisters were analysed for CHBr$_3$ and CH$_2$Br$_2$ and other trace compounds within 72 hours of collection. The WAS instrument was calibrated using the National Oceanic and Atmospheric
Administration (NOAA) 2003 scale for CHBr$_3$ and the NOAA 2004 scale for CH$_2$Br$_2$ (Jones et al., 2011; Andrews et al., 2016). For CONTRAST, a similar WAS system was employed to collect CHBr$_3$ and CH$_2$Br$_2$ measurements on the NSF/NCAR Gulfstream-V HIAPER (High-performance Instrumented Airborne Platform for Environmental Research) aircraft. A working standard was used to regularly calibrate the samples, and the working standard was
calibrated using a series of dilutions of high concentration standards that are linked to National Institute of Standards and Technology standards. The mean absolute percentage



error for CHBr$_3$ and CH$_2$Br$_2$ measurements between 0−8 km is 7.7% and 2.2% between the two WAS systems and two accompanying GC/MS instruments, respectively.

To interpret these atmospheric data we use the GEOS-Chem global 3-D atmospheric chemistry transport model (v9.03, http://geos-chem.org). We drive the GEOS-Chem model using GEOS-FP meteorological fields, provided by the Global Modeling and Assimilation Office at NASA Goddard, with a horizontal resolution of 2$^o$ (latitude) X 2.5$^o$ (longitude). We use a tagged version of the model (Butler et al, 2016) in which the atmospheric chemistry is

linearized by using pre-computed OH and photolysis loss terms using the same version of the model but with a more complete description of HOx-NOx-Ox and bromine chemistry (Parrelle et al., 2012). Our 3-D OH fields are consistent with the observed methyl chloroform lifetime.  We find small (5%) adjustments to these OH fields do not significantly affect our analysis or conclusions (not shown). For the purpose of our calculations we pre-compute

these loss terms every three hours during the campaign. This tagged modelling approach greatly simplifies the calculation of the Jacobian matrix used by the inverse model to determine surface flux estimates, as described below. We have previously evaluated this version of the model using CHBr$_3$ and CH$_2$Br$_2$ mole fraction data from NOAA/ESRL (Butler et al, 2016), and showed a level of agreement with *in situ* observations that is comparable to

the ensemble of models reported by Hossaini et al (2016a).

We use *a priori* emissions of CHBr$_3$ and CH$_2$Br$_2$ from the Ordóñez et al (2012) inventory, which is based on the top-down methodology using aircraft observations from 1996 to 2006. This represents one of three commonly used inventories, which were recently evaluated in a

multi-model inter-comparison study (Hossaini et al, 2016a). Liang et al (2010) also employed a top-down methodology to infer CHBr$_3$ and CH$_2$Br$_2$ fluxes, but Ziska et al. (2013) inferred these fluxes from a database of surface ocean observations collected from 1989 to 2011. We find no single inventory is best at reproducing observations of both gases. Ordóñez et al (2012) assumed a linear relationship between tropical CHBr$_3$ and CH$_2$Br$_2$ emissions and

monthly fields of chlorophyll-a, a proxy for ocean biological activity, to help fill in the spatial and temporal gaps left by the aircraft data. This approach strongly links the distributions of these two gases in the *a priori* inventory, an assumption we examine below. We primarily use Ordóñez et al. (2012) but also show the results from other inventories. For our study period, these aggregated regional fluxes are 6.2±0.9x10$^8$ g/month and 0.9±0.2x10$^8$ g/month

for CHBr$_3$ and CH$_2$Br$_2$ over 130$^o$—155$^o$E and 0$^o$—12$^o$N, respectively.

Figure 1 shows the geographical regions considered in this study. We divide the world into 605 basis functions:  1) a nested domain of 600 grid-scale tagged regions over the tropical





western Pacific (105°—165°E, -15°—25°N); 2) a lateral boundary of 15° surrounding the
nested domain, described by four tagged regions; and 3) the rest of the world. We spin-up
the model using *a priori* inventories (Ordóñez et al., 2012) from July 1ˢᵗ 2013 to January 18ᵗʰ
2014, reducing the impact of initial conditions.

We use the MAP approach to infer $CHBr_3$ and $CH_2Br_2$ surface fluxes from atmospheric mole
fraction measurements taken by CAST and CONTRAST aircraft campaigns. We infer
regional monthly mean surface fluxes, *f*, of $CHBr_3$ and $CH_2Br_2$:

$$f_p^g(x) = f_0^g(x) + \sum_i c_i^g BF_i^g(x), \tag{1}$$

where superscript *g* denotes trace gas, and the subscripts 0 and *p* denote the *a priori* and *a
posteriori* state vector, respectively We describe the regional fluxes as a product of a basis
function set $BF_i^g(x)$, representing distributions of monthly mean fluxes of the study gases
over 605 pre-defined geographic regions (Figure 1) over the duration of the CAST and
CONTRAST aircraft experiments, and scalar coefficients $c_i^g$ that are fitted to the data.

We include all the coefficients $c_i^g$ for the pre-defined 605 basis functions into the state vector
**c** that describes the $CHBr_3$ and $CH_2Br_2$ fluxes, which we fit to the observations. We take into
account the uncertainty of the model spin-up by including a scaling factor into the state
vector to adjust the background (initial) field, assuming that the model describes the
background vertical structure over the study domain. As a result the state vector **c** has a
total of 606 elements. We optimally estimate the state vector **c** by minimizing the associated
cost function $J(\mathbf{c})$:

$$J(\mathbf{c}) = \frac{1}{2}[\mathbf{c} - \mathbf{c}_0]^T \mathbf{B}^{-1}[\mathbf{c} - \mathbf{c}_0] + \frac{1}{2}\left(\mathbf{y}_{obs} - H(\mathbf{c})\right)^T \mathbf{R}^{-1}\left(\mathbf{y}_{obs} - H(\mathbf{c})\right), \tag{2}$$

where the superscripts T and -1 denote the matrix transpose and inverse operations,
respectively; $\mathbf{c}_0$ represents the a priori estimates; and **B** represents the *a priori* error
covariance matrix. The measurement vector, including the CAST/CONTRAST $CHBr_3$ and
$CH_2Br_2$ mole fraction data, is denoted by $\mathbf{y}_{obs}$, and **R** is the measurement error covariance
matrix. The forward model *H* projects the state vector (scalar coefficients) into observation
space (3-D mole fractions), and includes the GEOS-Chem atmospheric chemistry and
transport model that is sampled at the time and location of each observation.

We assume a 60% uncertainty for fluxes within the nested domain and a 50% uncertainty for
fluxes in the lateral boundary and the rest of the world regions, guided by the discrepancy
between the top-down and bottom-up inventories and their limited spatial and temporal
variation. We also assume that the *a priori* errors within the nested domain are correlated
over a distance of 400 km, corresponding to approximately the width of two adjacent grid





boxes. We assume the initial conditions for the mole fractions have a 30% uncertainty. We assume individual observations of $CHBr_3$ and $CH_2Br_2$ have errors of 20% and 10%, respectively, and are uncorrelated. These conservative values are guided by an analysis of data collected from different instruments during CAST and CONTRAST (Andrews et al, 2016). We assume that the observation error covariance **R** is diagonal, which also includes

model error, such as the representation error and the errors in modelling atmospheric transport and chemistry processes, with an assumed value of 20%.

The Jacobian matrix describes the sensitivity of atmospheric $CHBr_3$ and $CH_2Br_2$ CAST and CONTRAST measurements to changes in geographical surface emissions and the initial

value on January 18[th] 2014. We construct it by scaling the tagged tracers originating from a specific geographical region by surface fluxes from that region.

To avoid negative flux estimates due to, for example, an uneven distribution of observations we use value-dependent prior uncertainties for grid point flux estimates. We assume a

functional form for the uncertainty of the flux coefficient $c_i$ (equation 1):

$$\sigma(c_i) = \begin{cases} 0.8, c_i > -0.6 \\ 0.8 - 2(-0.6 - c_i)e^{k(1.0+c_i)}, \ c_i < -0.6, \end{cases} \tag{3}$$

where k (=3) is a pre-chosen factor that defines the gradient of the uncertainty with respect to the change of $c_i$. Using this approach, the *a priori* uncertainty decreases rapidly towards zero when $c_i$ becomes smaller than -0.6 (i.e., when the flux estimate is smaller than 40% of

the *a priori*). We find that using different parameters (e.g. changing the threshold from -0.6 to -0.8) does not significantly change our flux estimates.

**3 Results**

Forward Model Analysis
Figure 2 shows that the model overestimates the $CHBr_3$ concentrations by 0.1—0.7 pptv at altitudes from 0.5 to 12.5 km, with the largest values near the surface that reflects errors in *a priori* ocean fluxes (Hossaini et al., 2016a; Butler et al, 2016). The model has reasonable skill at reproducing the mean observed vertical gradient (r=0.62) but has a positive model

bias of 0.46±0.39 pptv. We find that vertical variations in $CHBr_3$ are determined approximately equally by sources over the western Pacific study region (Figure 1) and by sources immediately outside of the nested domain and further afield (Butler et al, 2016). These contributions show different vertical structures. The contribution from fresher sources over the western Pacific has a steeper atmospheric lapse rate from the boundary layer to the



free troposphere than the air masses from neighbouring regions. Both contributions are approximately uniform above the free troposphere, with the exception of a peak at 10-12 km from the air being transported into the nested domain (Butler et al, 2016). These differences in vertical structure help the inversion system identify the origin of $CHBr_3$ at different vertical levels.


The model reproduces some of the observed $CH_2Br_2$ variation (r=0.38) but with a small mean bias (0.01±0.14 pptv). Figure 2 shows that the $CH_2Br_2$ source outside the nested domain represent more than 60% (0.7—0.9 pptv) of the values sampled over the western Pacific, and almost invariant with altitude. This is due to weaker surface emissions over the
western Pacific and the longer atmospheric lifetime of $CH_2Br_2$ compared to $CHBr_3$. Ocean emissions from the western Pacific and from the immediate neighbouring regions each contribute only 0.1—0.3 pptv to $CH_2Br_2$. This highlights the difficulties of inferring ocean fluxes of $CH_2Br_2$ only using atmospheric $CH_2Br_2$ data collected over the western Pacific and considering this region in isolation.


To examine model transport errors associated with using a relatively coarse model spatial resolution ($2°×2.5°$), we ran a short, high-resolution ($0.25°×0.3125°$) simulation of $CHBr_3$ over a limited spatial domain centred on the western Pacific and compared that against the CAST/CONTRAST data. We acknowledge that we could still miss rapid, sub-grid scale
convective events using this model that has a factor-of-eight improvement in spatial resolution. However, we find that differences between the two model runs are much smaller than the differences between the individual model runs and the observations (Figure 2), suggesting that our model bias is mostly a result of overestimating ocean sources.

Closed-Loop Numerical Experiments
In the absence of independent observations to evaluate our *a posteriori* ocean fluxes we use closed-loop numerical experiments to understand what we can theoretically achieve from CAST and CONTRAST data, accounting for a realistic description of model and measurement errors.  These calculations, often called observing system simulation
experiment (OSSEs), provide an upper boundary on the ability of available data to infer the true state.

First, we generate synthetic observations at the time and location of the CAST and CONTRAST data by sampling 3-D model fields of $CHBr_3$ and $CH_2Br_2$ mole fraction driven by
the *a priori* inventories, which we regard as the 'true' emissions. We consider these sample



mole fraction values as the instrument observation after we superimpose instrument (unbiased) noise, informed by realistic observation uncertainty. Second, we enlarge the ('true') *a priori* emissions to generate the *a priori* estimate for the OSSEs: by 50% for emissions over the western Pacific and by 30% for emissions from the neighbouring region.

The resulting atmospheric mole fractions represent our model *a priori* concentrations. With perfect coverage of the atmosphere with perfect data (i.e. infinitesimal noise levels) fitting model emissions to the true observations would result in estimating the true ocean emissions. We describe our results as the difference between the *a posteriori* and true fluxes using a metric (Palmer et al, 2000; Feng et al, 2009) that describes the error reduction

$\gamma = 1 - \sigma_a/\sigma_f$, where $\sigma_a$ and $\sigma_f$ denote the *a posteriori* and *a priori* uncertainties, respectively, ignoring the correlation between state vector elements. The closer the value of $\gamma$ is to unity the larger reduction in uncertainty.

Figure 3 shows that the CAST/CONTRAST $CH_2Br_2$ and $CHBr_3$ measurements can

reproduce the true fluxes, mainly between $135\text{-}155^{\circ}E$ and $3^{\circ}S\text{-}15^{\circ}N$, by reducing the inflated *a priori* flux estimate. *A posteriori* fluxes in several grid boxes are lower than the true value, which is a result of regions overcompensating for other regions that have insufficient data to estimate their emissions. Regions influenced with fewer measurements (Figure 1) generally have smaller reductions in error, as expected. The error reductions for $CHBr_3$ range from 0.1

to 0.6 over the study domain, reflecting the widespread sensitivity of the CAST and CONTRAST observations to emissions from the tropical western Pacific region. The mean and median *a posteriori* fluxes are approximately a factor of three closer than the *a priori* to the true fluxes, with a 40% improvement in the uncertainties. In contrast, for $CH_2Br_2$, the error reduction is much smaller, with values greater than 0.3 only over a small geographical

region where the data density is greatest. There is a factor-of-two improvement in the discrepancy of the fluxes with the 'true', and a 30% improvement in the uncertainties. This large improvement in the knowledge of flux estimates is partly due to our simple description of the difference between the true and *a priori* field.

Ocean Emissions of $CHBr_3$ and $CH_2Br_2$ Inferred from CAST/CONTRAST data
We now examine the fluxes inferred from the CAST and CONTRAST measurements. Figure 4 shows elevated *a posteriori* $CHBr_3$ emissions surrounding small islands north of the tropics, such as Palau ($7.4^{\circ}N$, $134.5^{\circ}E$) and Chuuk ($7^{\circ}\,25'\,N$, $151^{\circ}\,47'\,E$). However, we find that emissions surrounding Guam ($13.5^{\circ}N$, $144.8^{\circ}E$) are not significantly different from

the adjacent open ocean. This reflects the distribution of boundary layer measurements (altitudes <2.5 km) of $CHBr_3$ observed during CAST and CONTRAST flights (Figure 1). We




find that through sensitivity experiments (not shown) that the *a posteriori* emissions are inferred by data and not via spatial correlations in the *a priori* emission inventory. Our *a posteriori* $CHBr_3$ emissions are generally higher than the bottom-up estimates from Ziska et al (2013), particularly over north of tropics.


We find that our *a posteriori* $CH_2Br_2$ emission estimates are lower than *a priori* estimates over open oceans north of $5^oN$. We also find elevated fluxes around islands and part of open oceans south of $5^oN$. Similar to $CHBr_3$, these elevated fluxes coincide with large boundary

layer measurements from CAST and CONTRAST.

Over the study domain ($130^o$—$155^oE$ and $0^o$—$12^oN$) our *a posteriori* fluxes are $3.6\pm0.3 \times 10^8$ g/month and $0.7\pm0.1 \times 10^8$ g/month for $CHBr_3$ and $CH_2Br_2$, respectively. These represent reductions of 40% and 20% relative to the *a priori* values, respectively. We find that our flux

estimates are largely insensitive to small changes in the assumed observation and *a priori* errors (not shown). The corresponding *a posteriori* mole fractions of $CHBr_3$ and $CH_2Br_2$ (not shown) have smaller mean biases ($-0.03\pm0.22$, $-0.1\pm0.11$) and improved correlations (r=0.74, r=0.56) than the *a priori* values compared to the observations.

The spatial gradient we find in our *a posteriori* $CHBr_3$ emissions between the coasts of Palaua and Chuuk and the surrounding open oceans is not present in our *a priori* emission inventory (Ordóñez et al, 2012). It is, however, qualitatively consistent with observations (e.g., O'Brien et al., 2009; Quack et al., 2007) and bottom-up estimates (e.g., Ziska et al., 2013 and Simmerler et al., 2015). These elevated coastal emissions also improve the fit to

CAST and CONTRAST observation particularly between 6-10 km. Figure 4 shows that the spatial distribution of *a priori* and *a posteriori* $CH_2Br_2$ emissions from the open ocean is different from the climatological bottom-up emissions (Ziska et al, 2013), particularly south of $5^oN$. This is surprising because studies have shown that tropical ocean emissions of $CH_2Br_2$ are correlated with the distribution of chlorophyll-a (e.g., Liu et al., 2013), but differences

may reflect inter-annual changes in ocean biology.

Figure 5 shows the *a priori* and *a posteriori* $CHBr_3$:$CH_2Br_2$ flux ratios. The top down inventory of Ordóñez et al (2012) use a linear model to describe emissions from these two gases, but the bottom-up inventory by Ziska et al, (2013) develop the emissions independently using a

database of ocean observations. This discrepancy between the two inventories is why we chose not to exploit this linear relationship in our MAP inversion. Our *a posteriori* emissions for $CHBr_3$ and $CH_2Br_2$ appear to be linearly related at low emissions but larger values appear



to follow a more complicated relationship, which may reflect differences in the responsible ocean biological processes.


## 4. Summary and Concluding Remarks

Very short-lived brominated gases have predominately natural sources, and therefore cannot be regulated by international agreements (Oman et al., 2016; Butler et al., 2007).
Current understanding of these natural sources is poor due to the infrequent and incomplete measurements of ocean fluxes that vary in space and time. Past studies have relied on developing bottom up inventories using a database of ship-borne measurements or an heuristic top down method that adjusted *a priori* emissions to match tropospheric and lower stratospheric measurements of a range of gases, including of $CHBr_3$ and $CH_2Br_2$. As a
consequence of the uncertainties associated with the modelling and data, the resulting inventories adopt simple distributions and are not necessarily consistent with each other on regional spatial scales.

Here, we used an *a priori* inventory to reproduce observed atmospheric boundary layer
variations of $CHBr_3$ and $CH_2Br_2$ over a small geographical region encompassing Guam, Pilau and Chuuk over the western Pacific. The measurements were collected as part of the CAST and CONTRAST aircraft campaigns during January and February 2014. We use the GEOS-Chem atmospheric chemistry model to relate the *a priori* emissions to the atmospheric concentrations, and develop a MAP inverse model to infer the ocean fluxes that
correspond with the aircraft measurements.

First, using a small number of closed-loop numerical experiments we showed that the aircraft data could in theory, using assumptions about their uncertainties, improve knowledge of ocean fluxes. Improvements in knowledge are generally related to the density of
measurements, as expected.

Using the aircraft data we find substantial spatial variations in fluxes of both gases that differ significantly from the *a priori* inventory. We find that aggregated regional *a posteriori* fluxes of $CHBr_3$ ($3.6\pm0.3\times10^8$ g/month) and $CH_2Br_2$ ($0.7\pm0.1\times10^8$ g/month) are 40% and 20% lower
than the *a priori* fluxes over the study domain ($130^o$—$155^o$E and $0^o$—$12^o$N). Using the model we find that observed variations of $CHBr_3$ are determined mainly by the open ocean while $CH_2Br_2$ has a large influence from outside the immediate study region. *A posteriori* fluxes significantly improve the mean observed vertical gradient of both gases, particularly in the free troposphere. We also find no evidence to suggest a robust linear relationship between



the emissions of these two gases over the study region, unlike one of the top-down *a priori*
inventories. This discrepancy may reflect differences in the analysis of data over different
spatial scales, or the construction of the *a priori* inventory using data in the free and upper
troposphere where observed air masses originating from disparate surface sources have
time to mix.


The MAP approach we used fits *a posteriori* fluxes to minimize the discrepancy between
model and observed atmospheric mole fractions. Any discrepancy in atmospheric data may
result from errors in surface fluxes (emissions minus uptake), atmospheric chemistry, and
atmospheric transport. The next most likely source of error is atmospheric transport,
particularly sub-grid scale vertical mixing. Sensitivity tests that crudely account for model
errors suggest that the *a posteriori* fluxes are robust.

Our paper highlights the value of using atmospheric data to improve the magnitude and
distribution of ocean emissions of halogenated gases, but also shows some of the difficulties
associated with interpreting these data even with the aid of an atmospheric transport model.
Future scientific progress in quantitatively understanding the role of natural emissions of
halogens in the catalytic destruction of stratospheric ozone is hampered by the lack of
available observations.

**Author contributions.**
L.F, P.I.P and R.B designed the computational experiments; P.I.P. and L.F. wrote the paper;
all authors provided input on data analysis shown in the paper; the CAST and CONTRAST
team provided access to CHBr$_3$ and CH$_2$Br$_2$ data.

**Acknowledgements**
L.F. was funded by United Kingdom Natural Environmental Research Council (NERC) grant
NE/J006203/1, R.B. was funded by NERC studentship NE/1528818/1, and P.I.P. gratefully
acknowledges his Royal Society Wolfson Research Merit Award. CAST is funded by NERC
and STFC, with grants NE/ I030054/1 (lead award), NE/J006262/1, 472 NE/J006238/1,
NE/J006181/1, NE/J006211/1, NE/J006061/1, NE/J006157/1,NE/J006203/1, NE/J00619X/1
(University of York CAST measurements), and NE/J006173/1. We are grateful to the
Harvard University GEOS-Chem group who maintains the model. E.A. acknowledges
support from NSF Grant AGS1261689 and thanks R. Lueb, R. Hendershot, X. Zhu, M.
Navarro, and L. Pope for technical and engineering support. The CONTRAST experiment is
sponsored by the NSF. CONTRAST data are publicly available for all researchers and can
be obtained at http://data.eol.ucar.edu/master_list/?project=CONTRAST. The NOAA surface
data is available at http://www.esrl.noaa.gov/gmd/dv/ftpdata.html.



**Figures**

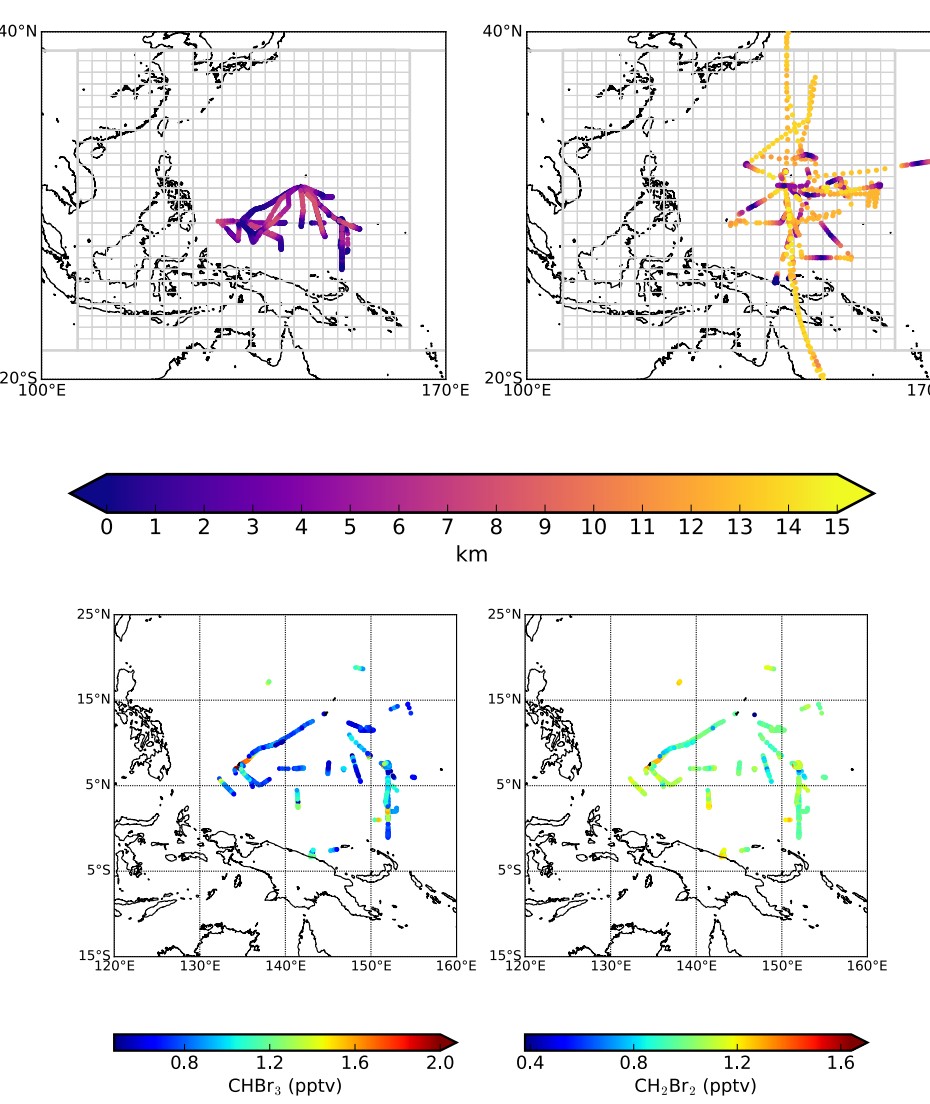


**Figure 1** Distributions of data from the (left) CAST and (right) CONTRAST aircraft campaigns during January and February 2014. Data are described on 2$^o$ (latitude) X 2.5$^o$ (longitude) GEOS-Chem grid boxes. The top panels show the altitude of data collected by both campaigns. We superimpose the flux inversion domain (grey lattice), consisting of 600

grid boxes between 105$^o$—165$^o$E and -15$^o$—25$^o$N, four larger neighbouring regions, and the rest of world. The bottom panels show the distributions of boundary layer (less than 2.5 km) CHBr$_3$ (pptv) and CH$_2$Br$_2$ (pptv) mole fraction data.






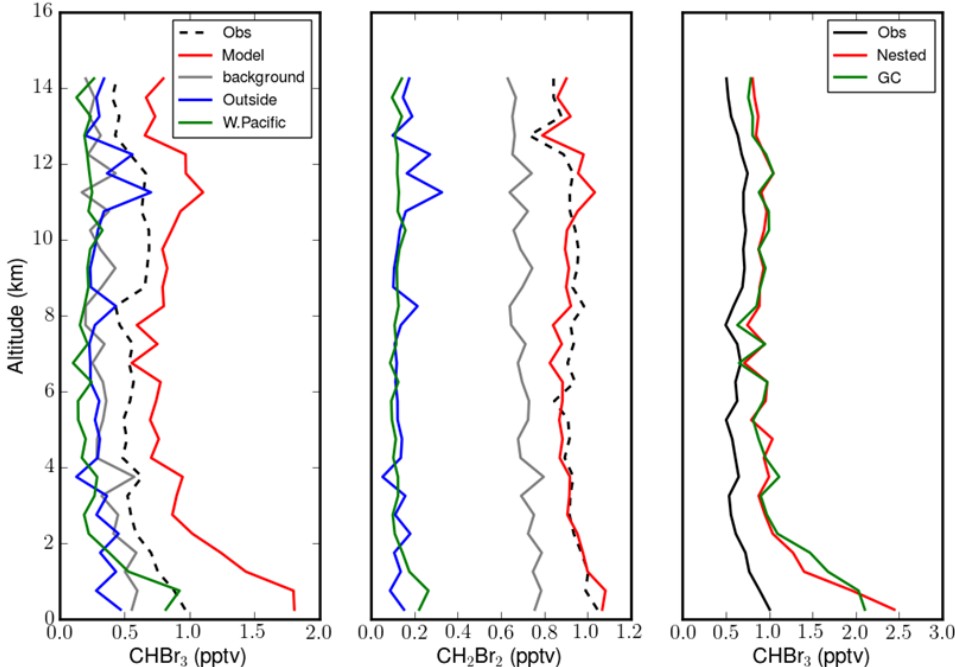

**Figure 2**: Observed and model mean vertical profiles of (left) CHBr$_3$ (pptv) and (middle) CH$_2$Br$_2$ (pptv) from the CAST and CONTRAST campaigns, described on a 1 km resolution grid. Model values have been sampled at the time and location of each observation. Also shown are the model contributions to these gases from within the Western Pacific study region, immediately outside the study region, and further afield which we denote as background values. The right panel compares CAST/CONTRAST observations of CHBr$_3$ with GEOS-Chem model simulations using the standard (2.0°×2.5°) and nested (0.25°×0.3125°) spatial resolutions from January 18[th] to February 13[th], 2014. Both model runs use common emission inventories.







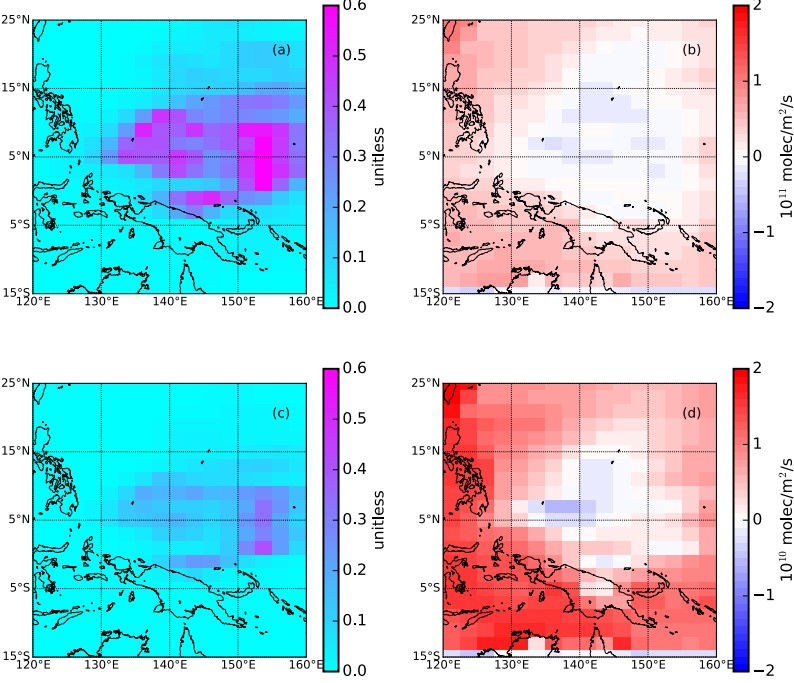

**Figure 3**: Simulated error reductions (unitless) and *a posteriori* flux error distributions ($10^{10}$ molec/m$^2$/s) of (top) CHBr$_3$ and (bottom) CH$_2$Br$_2$ based on the theoretical potential to recover true fluxes using the time and location of CAST and CONTRAST data.






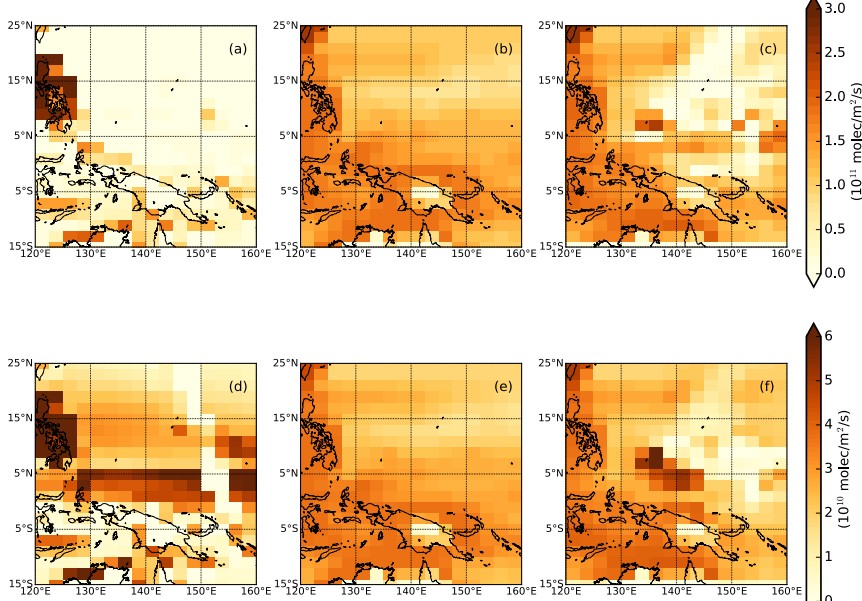

**Figure 4**: *A priori* and *a posteriori* (top) CHBr$_3$ (10$^{11}$ molec/m$^2$/s) and (bottom) CH$_2$Br$_2$ (10$^{10}$ molec/m$^2$/s) surface fluxes over the Western Pacific study region. The left panels show the *a priori* fluxes we use in our MAP inversion (Ordóñez et al., 2012); the middle panels shows an alternative bottom-up emission inventory (Ziska et al, 2013); and the right panels show our *a posteriori* flux estimates inferred from CAST and CONTRAST boundary layer data.





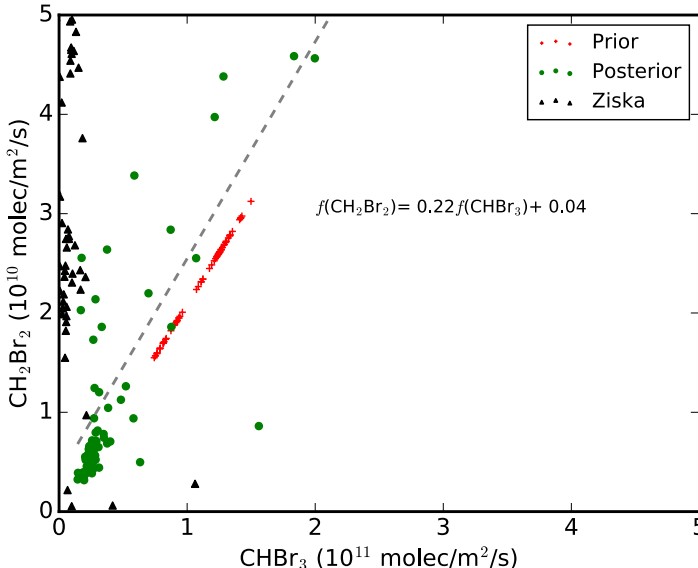

**Figure 5**: Scatterplot between *a priori* and *a posteriori* $CHBr_3$ and $CH_2Br_2$ fluxes described on $2^o$ (latitude) X $2.5^o$ (longitude) grid boxes over a sub-region ($130^o$—$155^o$E and $0^o$—$12^o$N) of the study region (Figure 1), where observations are most dense. Red crosses represent values from Ordóñez et al., 2012 that we use for our *a priori*; black triangles represent values from an alternative bottom-up inventory (Ziska et al, 2013); and green circles denote our *a posteriori* values. *A posteriori* fluxes of $CHBr_3$ and $CH_2Br_2$ have a Pearson correlation of 0.86. The best-fit linear model for the a posteriori fluxes is shown inset.



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
