# Peer review of "Surface fluxes of bromoform and dibromomethane over the tropical western Pacific inferred from airborne in situ measurements"

_Atmospheric Chemistry and Physics, 2017_

## Referee Comment (RC1) · Anonymous Referee #1 · 25 Jan 2018

Comments to:

**Surface fluxes of bromoform and dibromomethane over the tropical western Pacific inferred from airborne in situ measurements**

Liang Feng[1,2], Paul I. Palmer[1,2], Robyn Butler[2], Stephen J. Andrews[3], Elliot L. Atlas[4], Lucy J. 5 Carpenter[3], Valeria Donets[4], Neil R. P. Harris[5], Ross J. Salawitch[6], Laura L. Pan[7], Sue M. Schauffler[7]

Submitted to ACP

This manuscript explores an important topic regarding the recovery of the stratospheric ozone layer, in particular the emission rates of halogenated very short-lived substances (VSLS) at the surface. These compounds can be rapidly transported into the stratosphere, especially over areas of active, deep convection, and affect stratospheric ozone levels, thereby delaying the recovery process. Very few observations exist in key areas of the world where VSLS emissions and transport rates can be significant to the stratospheric budget of inorganic bromine, so the availability and use of new aircraft data sets constitutes a unique opportunity to test our models and evaluate our inventories.

I will first provide some general comments to the manuscript, followed by some specific ones.

General

1. While the topic is of great relevance, it is unclear what the contribution of the manuscript is, as written. Similar flux calculations were done by previous studies (lines 171-173). Is the contribution of this manuscript related to the methodology used, to the new aircraft data set over the tropical Western Pacific, and/or to the new magnitudes of fluxes obtained in this study?
2. The results presented in the manuscript are based on numerous model assumptions (e.g., lines 215 – 226). Were any sensitivity tests performed on the choice of values used? Are there any references to justify the choice of values used? Paragraphs 4 and 5 in the Introduction highlight how previous studies were based on several (different) assumptions and how those results need to be examined with caution. How can the results from this analysis, along with the assumptions used, be compared against previous studies?
3. The type of correlation between bromoform and dibromomethane is of importance in this analysis. What is the rationale for a linear correlation used in several studies published earlier (e.g., lines 103-105)? Given that the new aircraft data set elucidates a different correlation between the two

compounds, elaborating some more on this topic will highlight one of the new findings from this study.

4. There are several instances of missing punctuation marks such as commas and periods throughout the manuscript.

5. Some of the listed uncertainties are significant (e.g., line 262, line 342). What is the impact of these uncertainties on the conclusions of this study?

Specific

6. Abstract, line 25: An r value of 0.38 does not really qualify as "reasonably consistent" correlation.

7. Abstract, line 36: Which *a priori* inventory was used for the comparison?

8. Introduction, line 47: "The wide range of … lifetimes allows for …"

9. Introduction, line 55: "There is a wide range of…"

10. Data, paragraph 1: Are there any references available for the CAST and CONTRAST instruments?

11. Data, lines 146-149: What are "mean absolute percentage errors"? Which data set is higher? Are the differences uniform with height? How is this metric used in the analysis and how does it impact the results?

12. Data, lines 146-149: The second half of the statement is confusing as stated. WAS refers to the collection method and GC/MS to the analysis technique. Each campaign provided one data set. It might be simpler to state "…between the CAST and CONTRAST instruments", instead.

13. Data, lines 157-158: Is there a reference available to support the statement?

14. Data, lines 163-164: Are the referenced data from NOAA's ground network collected at the surface? Given that this study examines data at higher altitudes as well, are there any model comparisons with data at higher altitudes?

15. Data, line 186-187: Is a 6-month spin-up enough time and seasonally appropriate?

16. Results, line 248: Even with higher a priori ocean fluxes, the model still depletes bromoform much faster between the surface and 2 km than the observations show. Is this a result of chemistry, transport, and/or something else within the model?

17. Results, line 274-278: The right panel of Figure 2 shows that the model's vertical distribution of bromoform is practically the same when run at coarse and fine spatial resolutions. This suggests that sub-grid convection, assuming that the model resolves some events at the finer scale used, does not play a significant role in the modeled vertical profile. Is this result expected for a tracer with a relatively short lifetime and over a region of active, deep convection?

18. Results, line 293-294: How were the 50% and 30% chosen? How sensitive are the results of the analysis to these percentage choices?

19. Figure 1, line 445: Suggest using "$15^{o}S – 25^{o}N$"

20. Figure 1 and Figure 2: Are the in situ data shown in these figures an average of both aircraft data sets?

---

## Referee Comment (RC2) · Anonymous Referee #2 · 26 Jan 2018

Review of Feng et al. 'Surface fluxes of bromoform and dibromomethane over the tropical western Pacific inferred from airborne in situ measurements'

General comments:

This manuscript describes an inverse modelling study designed to infer biogenic, oceanic emissions of two short-lived bromocarbons (bromoform and dibromomethane- $CHBr_3$ and $CH_2Br_2$). The authors use an emission inversion setup consisting of a global chemistry transport model, a priori emission inventories for $CHBr_3$ and $CH_2Br_2$, and aircraft measurements from two separate campaigns measuring both compounds over the tropical Western Pacific. The authors also carry out a short observing system simulation experiment to retrieve a set of known idealised emissions as a means of proving the efficacy of their inversion modelling setup. The authors conclude that the a priori emissions of $CHBr_3$ and $CH_2Br_2$ are too high over this region, and find the a posteriori emissions of $CHBr_3$ and $CH_2Br_2$ to be lower than the a priori emissions by 40% and 20%, respectively. They also conclude that assumptions in previous studies of a correlation in the emissions of $CHBr_3$ and $CH_2Br_2$ cannot be supported based on the findings of this work.

The subject matter of the article sits well within the frame of ACP. In addition, the objectives of the scientific study and the design of the experiment (on the whole) mean that this work provides a useful scientific contribution on a topic (i.e. the biogenic, oceanic emissions of bromocarbons) where we have relatively poor understanding. It is welcome to see studies moving away from heuristic methods for inferring $CHBr_3$ and $CH_2Br_2$ emissions towards using more robust methods. I therefore find that this paper is a welcome and much needed scientific contribution.

Overall, I find the article to be well written and organised. The scientific ideas are laid out in a clear and logical manner, and consequently one can follow the flow of ideas easily. The article also does quite well at justifying the methodological choices, although there is one major issue here that I will highlight below in the specific comments section. Unfortunately, this issue does have a direct bearing on my recommendation for publication. Separately, and as a more minor issue, I did find that the authors stopped short somewhat of some deeper discussion that I feel would help strengthen the article, and I will explain this in more detail below.

I therefore find that this has the potential to be a very good scientific article. However, I cannot recommend publication until the issues outlined in the specific comments section are addressed.

Specific comments:

1) The most significant problem I find in this study relates to the choice of Ordonez et al as the sole choice for priori emission inventory for both $CH_2Br_2$ and $CHBr_3$ emissions.
   a. My first point relates to the $CHBr_3$ a priori emissions. I fully recognise the challenges Ordonez et al. faced in creating an emission inventory using heuristic methods in a global model, and I fully respect the useful contribution Ordonez et al. have made to our understanding of VSLS emissions. However, we now have several studies (including this study and Ordonez et al. itself) that show that the Ordonez et al. $CHBr_3$ emissions in particular are over estimated in the Western Pacific region.
      i. In fact Ordonez et al. (2012, ACP) itself in Figure 7 (the PEM-Tropics A, PEM-Tropics B, and TRACE-P panels) shows that their own model over estimates $CHBr_3$ in the Western Pacific region when using their own emissions.
      ii. Ashfold et al. (2014, ACP) - another study employing a top-down method to infer VSLS emissions in the tropical Western Pacific - derived lower estimates of $CHBr_3$ emissions in the tropics than Ordonez et al. Similarly, their retrieved

emissions (Fig. 6 panel D-F) show generally lower emission values than Ordonez et al., and the a posteriori emissions in this study (albeit in area not influenced by the emission inversion), in an overlapping region south of the Philippines.

iii. Hossaini et al. (2013, ACP) show the modelled tropical CHBr3 concentrations from the Ordonez et al emissions to be a consistently high outlier compared to those from the other emission data sets and observations (Fig.5 MLO, KUM, and SMO panels for the tropics; Fig. 7 30°N-30°S panels HIPPOS 1-5; and Fig. 11 all panels).

iv. Hossaini et al. (2016, ACP) Figs. 6 and 7 show that while using a much larger number of models, that the Ziska et al. emissions for CHBr3 generally outperform the Ordonez et al emissions in tropical locations.

v. In this study, the resulting simulated atmospheric concentrations of CHBr3 from the GEOS-CHEM CTM have a high bias as a result of using Ordonez et al. as the a priori.

vi. Unpublished work modelling seen by this reviewer that was presented at SHIVA meetings showing other models to overestimate CHBr3 concentrations in the western Pacific region when using the CHBr3 Ordonez et al emissions.

The conclusion I am making is not that the Ordonez et al emissions are too high in all regions of the world, but the preponderance of the evidence shows that they are too high over the important Western Tropical Pacific region considered in this study. I realise that the Ordonez emissions have been used recently by co-authors of this paper, and the link they have to ocean chlorophyll seems attractive, so perhaps they were a natural choice. However, based on the extensive evidence I presented above, I think that they were a poor choice (being the sole a priori tested). The a priori is an essential component of equation 2 necessary for a convergence to the global minimum of the cost function and for the best possible estimation of the emission state **c**. Therefore, selecting the best possible a priori emission dataset without large flaws is important for this study.

b. As a second point related to the first, I do not follow the logic of using the same published source for emission inventories of both CH2Br2 and CHBr3. One of the clear conclusions of the only comprehensive emission intercomparison study to date (Hossaini et al., 2013) was that no single emission inventory study was successful at producing good emissions for both compounds in question. The conclusions of Hossaini et al., were that Liang et al. provided the best estimates of CH2Br2 emissions over this region, and that Ziska et al. provided the best estimates of CHBr3 emissions. I do not think the authors should constrain themselves by picking the a priori emissions for two different compounds from the same published source when we know already that none of the published sources is able to satisfactorily give good results for both species.

c. I do have a further concern stemming from the fact that the a priori emission for CHBr3 seems to be too high and that the results of the OSSE show that the emission inversion setup has a tendency to overcompensate locally (to the observations). Simultaneously, the emission inversion seems to fail to significantly reduce errors in areas further away from the well observed areas of the domain. For the actual emission inversion, it seems entirely plausible that emissions could be being overly reduced in the well-observed area of the domain while remaining too high at the

western, northern and southern fringes. Can the authors please discuss how they think this issue affects their results.

Concluding my remarks on point 1), I strongly recommend, and as a condition of acceptance for publication, that in addition to running the emission inversion with Ordonez et al. as the a priori for both compounds, that the authors also run their emission inversion algorithm with Ziska et al. (CHBr3) and Liang et al. (CH2Br2) for the two compounds. Comparing this work to that of Ashfold et al. (2014, ACP), one can see that Ashfold et al. (2014, ACP) undertook a variety of emission inversion experiments (including changing their a priori) to test the setup of their system. These aspects of Ashfold et al. (2014, ACP) strengthened their work, and, similarly, this manuscript would also benefit from a similar effort.

2) Some key discussions seem to be missing including those of limitations of this study.
   a. It would have been nice to see a discussion of the prevailing meteorology during the period of study and an explanation linking this to the error reductions that we see in the OSSE results in Figure 3. Presumably, the error reductions are a function of the location of the observations and the origin of air masses arriving at the observation locations. An analysis similar to what I am suggesting was carried out in Ashfold et al. (2014) in their Figure 2, which allowed them to determine where there inversion setup was able to retrieve emission values. I realise this is perhaps easier in the Lagrangian framework of NAME, but the authors could draw upon the information in their meteorological inputs for GEOS-Chem to create a climatology of the winds and then make a discussion that would add useful context to the results and strengthen the paper.
   b. It would be good to see the authors try to connect the results of the OSSE, i.e., the spatially limited pattern of the error reduction, to the areas in the a posteriori CHBr3 emissions where we see the largest reductions in absolute emission values relative to the a priori. Given the evidence I present in point 1) above, I do not believe that the similarity in the spatial patterns in the OSSE error reduction and the area of reduced a posteriori emissions is coincidental. I think this implies that with greater spatial coverage in the aircraft observations that we would see reductions in the a posteriori emissions covering a larger spatial area. The authors should discuss this point, and also conclude that the spatial extent of the aircraft observations provides a limitation for this study.
   c. Further to point b., I do not find the a posteriori CHBr3 emission estimates outside of the region sampled by the observations (towards the N,S, W fringes of the domain) to be credible in light of the large reductions we see in the a posteriori compared to the a priori over the most sampled region. I am working on the assumption that the emissions are spatially correlated. Perhaps some discussion of this point in the context of the previous studies (e.g., those highlighted above in point 1) would help readers gauge the quality of the emission inversion in the areas on the N,S, W extremes of the domain where there is little information from the observations. This might also help readers understand the large gradients we see between the sampled and poorly sampled regions.
3) Figure colours in Figure 4 need greater differentiation. I struggled to differentiate the monochrome orange/brown tones. A set of panels representing the relative differences between the a priori and a posteriori emissions would also be of help.
4) I think it is necessary for the authors to include a discussion of the conclusions of Russo et al. (2015, ACP). Russo et al. (2015, ACP) made two conclusions relevant to the work in this study:

<ol type="a">
<li>That it is difficult to infer emissions using aircraft measurements and coarse global models in the case where the emission distribution is heterogeneous in regions of strong convective activity.</li>
<li>That model resolution can affect the simulated distributions of CHBr3 in cases where the emissions distribution is heterogeneous.</li>
</ol>

The authors should include some discussion of these points and should explain how they present limitations for the current work, or why this points are not relevant to the conclusions in this manuscript.

<ol start="5">
<li>It is important to note that the a posteriori emissions are more heterogeneous than the a priori. Therefore, following from Russo et al. (2015, ACP) and the discussions in point 4) above, the issue of model resolution could affect the simulated distribution of CHBr3 more significantly for the a posteriori emissions than for the a priori emissions. The authors have tested the impact of model resolution on the a priori emissions and found no effect. However, it seems plausible that model resolution could change the distributions of CHBr3 in the atmosphere more significantly for the a posteriori emissions given their greater heterogeneity. I recommend that the authors test this in a separate sensitivity study and present their conclusions.</li>
<li>It isn't clear to me that the mean bias between the mole fractions of observed and modelled CH2Br2 decrease from the a priori to the a posteriori simulations. The paper states this, but as it is written the bias changes from 0.01 +/- 0.14 to -0.1 +/-0.1. Can the authors please explain this result? Is this due to an overcompensation in the a posteriori emissions close the well observed region? According to the forward model section, a large fraction of the CH2Br2 originates from outside of the domain, and I imagine that in this case it is hard/impossible to infer those emissions with any reasonable specificity and overcompensation locally seems therefore to be a plausible explanation.</li>
</ol>

Technical comments:

Looking at Figure 4, it seems that the Ordonez et al and Ziska et al panels have been mislabelled in the caption whereby the Ziska emissions are described as being the Ordonez emissions and vice versa for the Ordonez emissions. Looking at Hossaini et al. (2013) ACP in figures 1 and 2 (and in fact the emission files themselves), I have checked the spatial patterns, and they seem to confirm this. Please can the authors check this themselves and confirm there is a mislabelling in the Fig. 4 caption? Please can the authors also check other instances of discussion of Ordonez and Ziska and verify that there a) there are no other mix-ups in the naming and b) that this is just a technical naming error.

Russo et al. (2015, ACP) is included as a reference but is not cited. Please check for other articles referenced but not cited.

---

## Author Comment (AC1) · 29 Jun 2018

**Replies to reviewer report 1:**

We thank the reviewer for the detailed and supportive comments. Below are our point-to-point replies.

**This manuscript explores an important topic regarding the recovery of the stratospheric ozone layer, in particular the emission rates of halogenated very short-lived substances (VSLS) at the surface. These compounds can be rapidly transported into the stratosphere, especially over areas of active, deep convection, and affect stratospheric ozone levels, thereby delaying the recovery process. very few observations exist in key areas of the world where VSLS emissions and transport rates can be significant to the stratospheric budget of inorganic bromine, so the availability and use of new aircraft data sets constitutes a unique opportunity to test our models and evaluate our inventories. I will first provide some general comments to the manuscript, followed by some specific ones.**

We thank the reviewer for the insightful comments.

General
**1. While the topic is of great relevance, it is unclear what the contribution of the manuscript is, as written. Similar flux calculations were done by previous studies (lines 171-173). Is the contribution of this manuscript related to the methodology used, to the new aircraft data set over the tropical Western Pacific, and/or to the new magnitudes of fluxes obtained in this study?**

We have applied a MAP approach to inferred $CHBr_3$ and $CH_2Br_2$ fluxes over tropical Western Pacific from the new CAST / CONTRAST experiments. Currently there are large differences in the distribution and magnitude of between existing $CHBr_3$ /$CH_2Br_2$ inventories (see, for example, the new Figure 6).

To our knowledge, we are the first to use the MAP approach to infer $CHBr_3$ and $CH_2Br_2$ surface fluxes over open oceans in tropical Western Pacific region, supported by new data from the CAST/CONTRAST campaigns. Our posterior estimates consistently show systematic deviations from the three independent prior inventories (see new Figure 6). These results have now been emphasized in the discussions (Page 11).

**2. The results presented in the manuscript are based on numerous model assumptions (e.g., lines 215–226). Were any sensitivity tests performed on the choice of values used? Are there any references to justify the choice of values used? Paragraphs 4 and 5 in the Introduction highlight how previous studies were based On several (different) assumptions and how those results need to be examined with caution. How can the results from this analysis, along with the assumptions used, be compared against previous studies?**

The reviewer is correct that we do introduce several assumptions to help infer surface fluxes from aircraft measurements. As wit other top-down flux inversions, we assume prior knowledge and its uncertainty. In the revised manuscript, we include sensitivity tests to test these assumptions about the magnitude of uncertainty (Lines 223-235), and three different sets of prior fluxes (Figure 6). The results from these tests demonstrate the robustness of our $CHBr_3$ fluxes over the main study domain where

observation coverage is relatively dense. Our posterior model simulations at two different spatial resolutions (revised Figure 2) are also in better agreement with observations than those based on prior inventories. However more independent data including the direct flux measurements are needed to fully evaluate our results in particular for $CH_2Br_2$ fluxes (See discussion).

**3. The type of correlation between bromoform and dibromomethane is of importance. What is the rationale for a linear correlation used in several studies published earlier (e.g., lines 103-105)? Given that the new aircraft data set elucidates a different correlation between the two 2 compounds, elaborating some more on this topic will highlight one of the new findings from this study.**

Several previous studies have assumed the linear correlation between $CHBr_3$ and $CH_2Br_2$ based on some observations as well as on the assumption about the shared biogenic sources. However other measurements and model studies suggest a rather complicated correlation between these two species. Our inversions also show no evidence to support such a simple linear relation. But our posteriori flux uncertainties, in particular for $CH_2Br_2$ (Figure 3), are too large for us to reach a definitive conclusion.

**4. There are several instances of missing punctuation marks such as commas and periods throughout the manuscript.**

During the revision, we have corrected the punctuation.

**5. Some of the listed uncertainties are significant (e.g., line 262, line 342). What is the impact of these uncertainties on the conclusions of this study?**

Our current estimates, in particular for $CH_2Br_2$ still have large uncertainties, limited by the observation quality and coverage, as well as by transport model errors. We believe they will only be addressed by more coordinated fluxes/concentrations measurements.

**Specific**

6. **Abstract, line 25: An r value of 0.38 does not really qualify as "reasonably consistent" correlation.**

Here the consistence is about the agreement with vertical distributions. We have clarified the text.

**7. Abstract, line 36: which a priori inventory was used for the comparison?**
It is from Ordonez et al. (2012). We have clarified this in the abstract.

**8. Introduction, line 47: "The wide range of … lifetimes allows for …"**
Changed the sentence as suggested

**9. Introduction, line 55: "There is a wide range of…"**
Changed the sentence as suggested

**10. Data, paragraph 1: Are there any references available for the CAST and**

**CONTRAST instruments?**

See Andrews et al. (2016).

**11. Data, lines 146-149: What are "mean absolute percentage errors"? Which data set is higher? Are the differences uniform with height? How is this metric used in the analysis and how does it impact the results?**

Their deviations change with altitudes (see Andrews et al 2016). By design, our inversions depend on observed horizontal and vertical gradients in the boundary layer, mainly observed by CAST. Most CONTRAST measurements are at much higher altitudes and hence less sensitive to local sources.

Our sensitivity experiments ), in which we introduce a bias between CONTRAST and CAST data that we infer in our inversion, show very similar results to our control run. We have included this in the main text.

**12. Data, lines 146-149: The second half of the statement is confusing as stated. WAS refers to the collection method and GC/MS to the analysis technique. Each campaign provided one data set. It might be simpler to state "…between the CAST and CONTRAST instruments", instead**.

Good suggestion. We have changed accordingly.

**13. Data, line 157-158: Is there a reference available to support the statement?**

See Butler et al (2016)

**14. Data, lines 163-164: Are the referenced data from NOAA's ground network collected at the surface? Given that this study examines data at higher altitudes as well, are there any model comparisons with data at higher altitudes?**

Yes. They are NOAA surface measurements. Unfortunately we cannot find independent aircraft measurements to evaluate our prior or posterior model simulations.

**15. Data, line 186-187: Is a 6-month spin-up enough time and seasonally appropriate?**

It is appropriate because of the short lifetime (<4 months) of the species.   In the inversion, we are also only focused on January and  February, 2014.

**16. Results, line 248: Even with higher a priori ocean fluxes, the model still depletes bromoform much faster between the surface and 2 km than the observations show. Is this a result of chemistry, transport, and/or something else within the model?**

We believe the higher lapse rate is likely related to the issues with model vertical transport, as it has also been found for posterior model simulation (Figure 2) even at finer model resolution.

**17. Results, line 274-278: The right panel of Figure 2 shows that the model's vertical distribution of bromoform is practically the same when run at coarse and fine spatial resolutions. This suggests that sub-grid convection, assuming**

**that the model resolves some events at the finer scale used, does not play a significant role in the modeled vertical profile. Is this result expected for a tracer with a relatively short lifetime and over a region of active, deep convection?**

The coarse and fine model simulations show different atmospheric lapse rate in the boundary layer for both prior and posterior surface fluxes. Even with our fine-scale model simulation (spatial resolution of ~25 km) there are sub-grid scale processes that are unaccounted. The role of model error is the subject of ongoing work.

**18. Results, line 293-294: How were the 50% and 30% chosen? How sensitive are the results of the analysis to these percentage choices?**

These percentages are chosen just to demonstrate observation constraints. The error reduction is insensitive to these values. Also, as shown in Figure 6, our inversion results are not sensitive to *a priori* fluxes over regions with proper observation coverage.

**19. Figure 1, line 445: Suggest using "15°S–25°N"**
Thanks. We have changed the latitude range as suggested.

**20. Figure 1 and Figure 2: Are the in situ data shown in these figures an average of both aircraft data sets?**

Figure 1 shows the altitudes of the CAST/CONTRAST measurements, and Figure 2 shows the $CHBr_3$ concentrations at the boundary layer, which are mainly observed by CAST.

**Reviewer report 2**

We thank the reviewer for providing a second round of reviewer comments. Below are our point-to-point responses to the reviewers' comments (denoted in bold). In particular, following the reviewer's suggestion, we have conducted additional inversion experiments using Liang's inventories as our a priori, and include the results to the main text (Section 3 and Section 5). In short, we find that our posterior flux estimates are robust against using different prior emission inventories, subject to data coverage.

**General comments**

**This manuscript describes an inverse modelling study designed to infer biogenic, oceanic emissions of two short-lived bromocarbons (bromoform and dibromomethane- CHBr3 and CH2Br2). The authors use an emission inversion setup consisting of a global chemistry transport model, a priori emission inventories for CHBr3 and CH2Br2, and aircraft measurements from two separate campaigns measuring both compounds over the tropical Western Pacific. The authors also carry out a short observing system simulation experiment to retrieve a set of known idealised emissions as a means of proving the efficacy of their inversion modelling setup. The authors conclude that the a priori emissions of CHBr3 and CH2Br2 are too high over this region, and find the a posteriori emissions of CHBr3 and CH2Br2 to be lower than the a priori emissions by 40% and 20%, respectively. They also conclude that assumptions in previous studies of a correlation in the emissions of CHBr3 and CH2Br2 cannot be supported based on the findings of this work.**

**The subject matter of the article sits well within the frame of ACP. In addition, the objectives of the scientific study and the design of the experiment (on the whole) mean that this work provides a useful scientific contribution on a topic (i.e. the biogenic, oceanic emissions of bromocarbons) where we have relatively poor understanding. It is welcome to see studies moving away from heuristic methods for inferring CHBr3 and CH2Br2 emissions towards using more robust methods. I therefore find that this paper is a welcome and much needed scientific contribution.**

**Overall, I find the article to be well written and organised. The scientific ideas are laid out in a clear and logical manner, and consequently one can follow the flow of ideas easily. The article also does quite well at justifying the methodological choices, although there is one major issue here that I will highlight below in the specific comments section. Unfortunately, this issue does have a direct bearing on my recommendation for publication. Separately, and as a more minor issue, I did find that the authors stopped short somewhat of some deeper discussion that I feel would help strengthen the article, and I will explain this in more detail below.**

**I therefore find that this has the potential to be a very good scientific article. However, I cannot recommend publication until the issues outlined in the specific comments section are addressed.**

We thank the reviewer for their supportive comments. We originally chose the dibromomethane and bromoform ocean flux estimates from Ordóñez et al (2012). as our prior because they provided more detailed spatial patterns than Liang et al (2010). As we show in the new Figure 6, using two alternative prior inventories (Ziska et al (2013) and. Liang et al. (2010)) does not significantly impact our results subject

to coverage provided by the aircraft data

We broadly agree with previous studies (but for a larger geographical region) that the Ordóñez inventory overestimated bromoform emissions over the Western Tropical Pacific region by nearly 40%. In our study, we also found large differences between open ocean fluxes and coastal (island) sources, contrary to Ordóñez et al, and also different from the simple spatial pattern over open oceans suggested by the inventory from Liang et al. (2010).

However, we agree with the reviewer that additional inversions informed by alternative prior knowledge can further this study. We now present additional experiments that use Ziska's and Liang's prior bromomethane emission inventory. These are included in the main text but summarized below.

**Specific comments:**

*1) The most significant problem I find in this study relates to the choice of Ordonez et al as the sole choice for priori emission inventory for both CH2Br2 and CHBr3 emissions.*
*a. My first point relates to the CHBr3 a priori emissions. I fully recognise the challenges Ordonez et al. faced in creating an emission inventory using heuristic methods in a global model, and I fully respect the useful contribution Ordonez et al. have made to our understanding of VSLS emissions. However, we now have several studies (including this study and Ordonez et al. itself) that show that the Ordonez et al. CHBr3 emissions in particular are over estimated in the Western Pacific region.*
*i. In fact Ordonez et al. (2012, ACP) itself in Figure 7 (the PEM-Tropics A, PEM-Tropics B, and TRACE-P panels) shows that their own model over estimates CHBr3 in the Western Pacific region when using their own emissions.*
*ii. Ashfold et al. (2014, ACP) - another study employing a top-down method to infer VSLS emissions in the tropical Western Pacific - derived lower estimates of CHBr3 emissions in the tropics than Ordonez et al. Similarly, their retrieved western, northern and southern fringes. Can the authors please discuss how they think this issue affects their results.*

We agree that the aircraft observations do not uniformly cover the study domain, but some of these gaps are effectively filled by atmospheric mixing of surface sources. Our inversion system includes scaling factors not only for sources within the study region but also for neighbouring regions that lie outside our study regions and for the initial conditions at the beginning of the study period. Our sensitivity experiments reveal that our results are not sensitive to global priori inventory when the observation constraints are strong such as the for $CHBr_3$ emissions over the study domain between between 130°—155°E and 0°—12°N. This is discussed in section 4.

*3. Concluding my remarks on point 1), I strongly recommend, and as a condition of acceptance for publication, that in addition to running the emission inversion with Ordonez et al. as the a priori for both compounds, that the authors also run their emission inversion algorithm with Ziska et al. (CHBr3) and Liang et al. (CH2Br2) for the two compounds. Comparing this work to that of Ashfold et al. (2014, ACP), one can see that Ashfold et al. (2014, ACP) undertook a variety of emission inversion experiments (including changing their a priori) to test the setup of their system. These aspects of Ashfold et al. (2014, ACP) strengthened their work, and, similarly, this manuscript would also benefit from a similar*

*effort.*

Figure 1 summarizes our results from using different prior inventories. Posterior flux estimates of $CH_3Br$ over the geographical region covered by CAST/CONTRAST aircraft data are remarkably similar. This supports the idea that the data are playing a significant role in determining the posterior flux estimates.

[Figure]

**Figure 1:** (Left upper panels) Prior and (left lower panels) posterior $CHBr_3$ flux estimates ($10^{11}$ molec/m$^2$/s) over the study region. The three prior inventories include Liang et al (2010), Ordóñez et al (2012), and. Ziska et al (2013). The right panel is focused on the geographical region between ($130°$—$155°E$ and $0°$—$12°N$) between where the CAST/CONTRAST had the most information.

*2) Some key discussions seem to be missing including those of limitations of this study.*
*a. It would have been nice to see a discussion of the prevailing meteorology during the period of study and an explanation linking this to the error reductions that we see in the OSSE results in Figure 3. Presumably, the error reductions are a function of the location of the observations and the origin of air masses arriving at the observation locations. An analysis similar to what I am suggesting was carried out in Ashfold et al. (2014) in their Figure 2, which allowed them to determine where there inversion setup was able to retrieve emission values. I realise this is perhaps easier in the Lagrangian framework of NAME, but the authors could draw upon the information in their meteorological inputs for GEOS-Chem to create a climatology of the winds and then make a discussion that would add useful context to the results and strengthen the paper.*

We agree that it is of great interest to show the origins of airmass, which has been partially investigated by another study (Bulter et al, 2016). However, the complexity of the global CTM and its analyzed meteorological fields and the nature of the aircraft measurements precludes a simple and intuitive summary of the overall sensitivity of the CAST and CONTRAST observation to the underlying surface fluxes. In addition, we have included scaling factor for initial concentrations and for emissions from neighbouring regions, and as a result, the posterior flux estimates are more or less dependent on the difference between modelled and observed internal horizontal and vertical gradients, instead of single concentration values (which has also been

revealed by consistency in posterior fluxes when different a priori is used.

*b. It would be good to see the authors try to connect the results of the OSSE, i.e., the spatially limited pattern of the error reduction, to the areas in the a posteriori CHBr3 emissions where we see the largest reductions in absolute emission values relative to the a priori. Given the evidence I present in point 1) above, I do not believe that the similarity in the spatial patterns in the OSSE error reduction and the area of reduced a posteriori emissions is coincidental. I think this implies that with greater spatial coverage in the aircraft observations that we would see reductions in the a posteriori emissions covering a larger spatial area. The authors should discuss this point, and also conclude that the spatial extent of the aircraft observations provides a limitation for this study.*

This is an interesting point. Aircraft measurements are sensitive to a wider range of geographical regions than the error reduction suggests, e.g. Figure 2 and Butler et al, 2017. The inversion updates the ocean fluxes over a wide geographical domain but the error reduction is often small because of the low signal (contribution) to noise (observation error and model transport error) ratio.

*Further to point b., I do not find the a posteriori CHBr3 emission estimates outside of the region sampled by the observations (towards the N,S, W fringes of the domain) to be credible in light of the large reductions we see in the a posteriori compared to the a priori over the most sampled region. I am working on the assumption that the emissions are spatially correlated. Perhaps some discussion of this point in the context of the previous studies (e.g., those highlighted above in point 1) would help readers gauge the quality of the emission inversion in the areas on the N,S, W extremes of the domain where there is little information from the observations. This might also help readers understand the large gradients we see between the sampled and poorly sampled regions.*

See response to previous point.

*Figure colours in Figure 4 need greater differentiation. I struggled to differentiate the monochrome orange/brown tones. A set of panels representing the relative differences between the a priori and a posteriori emissions would also be of help.*

Agreed. The manuscript (Figure 4) has been amended, accordingly.

*4) I think it is necessary for the authors to include a discussion of the conclusions of Russo et al. (2015, ACP). Russo et al. (2015, ACP) made two conclusions relevant to the work in this study:*

*a) That it is difficult to infer emissions using aircraft measurements and coarse global models in the case where the emission distribution is heterogeneous in regions of strong convective activity.*
*b). That model resolution can affect the simulated distributions of CHBr3 in cases where the emissions distribution is heterogeneous.*
*The authors should include some discussion of these points and should explain how they present limitations for the current work, or why this points are not relevant to the conclusions in this manuscript.*

*The authors should include some discussion of these points and should explain how they present limitations for the current work, or why this points are not relevant to the conclusions in this manuscript.*

Certainly, using a finer-scale model resolution would be preferable. Our forward simulation at the native model resolution of 0.25° × 0.3125° confirm that posterior fluxes result in a better agreement with observations than the prior as shown in the revised Figure 2. However, the resolution of estimates fluxes is determined by the quality, quantity, and distribution of available data. In this case, even if we used a finer resolution model it is likely we would need to aggregate model grid values to generate estimates that do not include large spatial correlations. We have included this discussion in the revised manuscript, following the reviewer's recommendation (Page 11).

*5) It is important to note that the a posteriori emissions are more heterogeneous than the a priori. Therefore, following from Russo et al. (2015, ACP) and the discussions in point 4) above, the issue of model resolution could affect the simulated distribution of CHBr3 more significantly for the a posteriori emissions than for the a priori emissions. The authors have tested the impact of model resolution on the a priori emissions and found no effect. However, it seems plausible that model resolution could change the distributions of CHBr3 in the atmosphere more significantly for the a posteriori emissions given their greater heterogeneity. I recommend that the authors test this in a separate sensitivity study and present their conclusions.*

Good suggestion. We have included such a comparison the revised manuscript (at revised Figure 2), which confirms that the posterior nested simulation at 0.25°×0.3125° is very similar to the run at 2°×2.5°.

*6) It isn't clear to me that the mean bias between the mole fractions of observed and modelled CH2Br2 decrease from the a priori to the a posteriori simulations. The paper states this, but as it is written the bias changes from 0.01 +/- 0.14 to -0.1 +/-0.1. Can the authors please explain this result? Is this due to an overcompensation in the a posteriori emissions close the well observed region? According to the forward model section, a large fraction of the CH2Br2 originates from outside of the domain, and I imagine that in this case it is hard/impossible to infer those emissions with any reasonable specificity and overcompensation locally seems therefore to be a plausible explanation.*

We agree that it is due to an overcompensation caused by an uneven sensitivity (Jacobian) for measurements at different altitudes, and observation errors that are assumed to be proportional to mole fraction values by two campaigns. See Pages 10 and 11 for more discussions

*Technical comments:*
*Looking at Figure 4, it seems that the Ordonez et al and Ziska et al panels have been mislabelled in the caption whereby the Ziska emissions are described as being the Ordonez emissions and vice versa for the Ordonez emissions. Looking at Hossaini et al. (2013) ACP in figures 1 and 2 (and in fact the emission files themselves), I have checked the spatial patterns, and they seem to confirm this. Please can the authors check this themselves and confirm there is a mislabelling in the Fig. 4 caption? Please can the authors also check other instances of discussion of Ordonez and Ziska and verify that there a) there are no other mix-*

***ups in the naming and b) that this is just a technical naming error.***

We are grateful to the reader for spotting this error. We have checked and can confirm that it is just a plot labelling error.

***Russo et al. (2015, ACP) is included as a reference but is not cited. Please check for other articles referenced but not cited.***

Thanks. We have now checked the reference list and cite Russo et al in the revision (see above).

---

## Referee Report (RR1)

General comments

I am very happy with author responses and the changes made to the manuscript. The authors addressed my concerns regarding possible uncertainties resulting from the use of a single a priori by carrying out a sensitivity study whereby they repeated the emission inversion with two other published inventories for CHBr3. I now fully support publication of this manuscript following only minor changes.

Specific Comments

Page 7, lines 250-253. You should probably make it a little clearer here that the forward model was run using the Ordonez emissions. You have three a priori tests now, so it may not be 100% clear that these forward model results are attributable to those emissions.

I recommend that the authors please find a new way to write the caption for Figure 4. I think it could be improved. For example, as the sentences jump from middle, to left, and then right. The different panels are labelled a, b, c, etc. I would recommend that you use these labels to help the reader. Another possible way to improve it would be better labelling on the figures themselves, e.g., labelling of the rows as CHBr3 and CH2Br2, and labelling the columns Ziska a priori, Ordonez a priori, and a posteriori. Something similar has been done for Figure 6, but note the comments below.

Page 10, line 357. "we find significant improvement". Do you mean *insignificant*? This seems more consistent with earlier discussion, Figure 2, and the argumentation that follows this line.

Page 11, line 378. You mention "*changes in ocean biology*". It might be worth mentioning the link between ocean biology and climatic variability. See for instance Fig. 6 in Racault et al., 2017 "*Impact of El Niño Variability on Oceanic Phytoplankton*" in Frontiers in Marine Science.

Page 11, lines 380 onwards. There is possibly experimental support for the lack of a simplistic relationship between emissions of CHBr3 and CH2Br2 (at least for macro algae). Recommend reading Sect. 3.3. of Leedham et al., 2013 "Emission of atmospherically significant halocarbons by naturally occurring and farmed tropical macroalgae" in Biogeosciences. Leedham et al. demonstrates correlations between the emissions of these compounds for single species, but the ratio of emission varies wildly.

I found the text in the Figure 5 caption to be quite unclear. The caption states it is a plot between "*a priori and a posteriori CHBr3 and CH2Br2 fluxes*". I would recommend the authors to simplify the description to state it is a plot of X versus Y, i.e., in this case CH2Br2 fluxes versus CHBr3 fluxes. Then state the additional complexity that this plot includes both a priori (from two sources) and a posteriori data. As it is, too many things have been stated in a single sentence.

Technical Comments

Page 3, line 89. Please change "*The bottom-up approach assumes local flux estimates are representation of larger spatial scales.*" To "The bottom-up approach assumes local flux estimates are *representative* of larger spatial scales."

Page 6, line 194. Missing full stop after "respectively".

Page 7, line 244. Missing comma after "e.g."

Page 8, line 282. Remove "the" before "a posteriori".

Page 8, lines 280-284. Recommend changing the current text from "*are more consistent with the observations that a priori fluxes.*" to "are more consistent with the observations **than those from the** a priori fluxes."

Page 10, lines 334. Recommend changing "*particularly over north of tropics*" to "particularly over the north of tropics"

Page 10, line 336. Missing "the" before "a priori"

Page 10, line 339. "*coincide with large boundary layer measurements from CAST and CONTRAST.*". Measurements of what? It's perhaps better to be precise and state CH2Br2 given that the sentence starts by mentioning CHBr3.

Page 11, line 381. Change "use" to "uses".

Page 11, line 382. Change "develop" to "develops".

Figure 1. I would recommend that you label the individual panels with instructive headings such as CONTRAST and CAST.

Figure 2 caption. There is an unnecessary full stop in the second to last line of the caption.

Figure 3. The panels are labelled a, b, c, etc. but these labels are not used in the caption. I recommend that you use the labels to improve clarity.

Figure 6. I find the caption description to be unclear. To simplify this I would recommend re-designing the plot by shifting the three plots in the column on the right hand side to be a new row underneath the a posteriori row. Call this new row a posteriori zoom, for instance. I would recommend moving the "Posterior" label to above the bottom row in Figure 6 rather below it. In addition, The Ordonez label in Figure 6 is barely readable. I would recommend moving the labels for each column to be placed above the panels in black type font.

---

## Referee Report (RR2)

Comments for revised version (v4):

**Surface fluxes of bromoform and dibromomethane over the tropical western Pacific inferred from airborne in situ measurements**

Liang Feng[1,2], Paul I. Palmer[1,2], Robyn Butler[2], Stephen J. Andrews[3], Elliot L. Atlas[4], Lucy J. 5 Carpenter[3], Valeria Donets[4], Neil R. P. Harris[5], Ross J. Salawitch[6], Laura L. Pan[7], Sue M. Schauffler[7]

Submitted to ACP

This is a very good revised version of the manuscript. The authors made very effective changes and also added new key material that further supports this study. For instance, adding the posterior model simulations in Figure 2 clearly and nicely illustrates the improvements obtained when having and using observations. Another example is Figure 6, which shows the impact of the choice of the a priori on estimating a posteriori fluxes when observations are available.

In reviewing this new version, I am left with one unanswered question (or perhaps I missed the answer) and a few specific suggestions that I would like to offer.

Question

1- Why is the agreement between model and observations better for $CHBr_3$, which has a shorter lifetime, than for $CH_2Br_2$ (lines 24-26)?

Suggestions

1- The addition of the new material illustrated in Figure 6 was very interesting and provides a very important finding (lines 400-401) that should be mentioned in the abstract and in the conclusion, namely that the choice of a priori plays a small role in determining a posteriori fluxes when atmospheric measurements are combined with the framework presented in this study (Geos-Chem and MAP).
2- Line 130: as with many acronyms properly defined throughout the manuscript, "GC/MS" should be spelled out as well.
3- Lines 134-145: the two data sets (CAST and CONTRAST) are calibrated using different scales, as the authors described. Are there any differences in scales? If so, were they accounted for in this analysis? The authors mentioned an offset between the two data sets (line 360-362) that were found when comparing to model outputs. Could that be due to the difference in scales?

4- Lines 145-148: how were these errors determined – this work or Andrews *et al*? If the latter, then it should be referenced.

5- Line 351: "This model bias...". Which bias, the negative one? Clarify this statement.

6- Multi-panel figures: consider using labels (e.g., (a), (b), (c)) in the panels and then add them to the captions for more efficient referencing. In some cases, for instance, Fig 3 and Fig 4, the labels were present in the panels, but not added to the captions.

7- Figure 2: provide acronym in caption for "GC" legend in rightmost panel.

8- Figure 2: was the GC analysis for $CHBr_3$ also done for $CH_2Br_2$? If so, what were the results?

9- Figure 2: the left panel shows that the model has a positive bias at all altitudes. The middle panel, however, shows both positive and negative biases, granted they are smaller in magnitude. What are the uncertainties or error bars in these vertical profiles, which can help assess the relevance of the model-observation differences?

10- Figure 6: are similar results obtained when examining $CH_2Br_2$? A few sentences on those results could be added to the text without the need to create new panels or figures.

11- Figure 6: include axes labels (lats/lons) in plots on the right.

---

## Author Response (AR2)

**Review report 1**

*General comments*

*I am very happy with author responses and the changes made to the manuscript. The authors addressed my concerns regarding possible uncertainties resulting from the use of a single a priori by carrying out a sensitivity study whereby they repeated the emission inversion with two other published inventories for CHBr3. I now fully support publication of this manuscript following only minor changes.*

We are grateful for the reviewer's kind support and insightful comments. Our point-to-point responses (in red) are detailed below:

*Specific Comments*

*1. Page 7, lines 250-253. You should probably make it a little clearer here that the forward model was run using the Ordonez emissions. You have three a priori tests now, so it may not be 100% clear that these forward model results are attributable to those emissions.*

Agreed. We now add '… driven by emissions from Ordóñez et al. (2012)'

*2. I recommend that the authors please find a new way to write the caption for Figure 4. I think it could be improved. For example, as the sentences jump from middle, to left, and then right. The different panels are labelled a, b, c, etc. I would recommend that you use these labels to help the reader. Another possible way to improve it would be better labelling on the figures themselves, e.g., labelling of the rows as CHBr3 and CH2Br2, and labelling the columns Ziska a priori, Ordonez a priori, and a posteriori. Something similar has been done for Figure 6, but note the comments below.*

We change figure 4 and caption as suggested:

*3.Page 10, line 357. "we find significant improvement". Do you mean insignificant? This seems more consistent with earlier discussion, Figure 2, and the argumentation that follows this line.*

Agreed, We have changed the text.

4. *Page 11, line 378. You mention "changes in ocean biology". It might be worth mentioning the link between ocean biology and climatic variability. See for instance Fig. 6 in Racault et al., 2017 "Impact of El Niño Variability on Oceanic Phytoplankton" in Frontiers in Marine Science.*

We thank the reviewer for this good suggestion, and we add the citation.

5. *Page 11, lines 380 onwards. There is possibly experimental support for the lack of a simplistic relationship between emissions of CHBr3 and CH2Br2 (at least for macro algae). Recommend reading Sect. 3.3. of Leedham et al., 2013 "Emission of atmospherically significant halocarbons by naturally occurring and farmed tropical macroalgae" in Biogeosciences. Leedham et al. demonstrates correlations between the emissions of these compounds for single species, but the ratio of emission varies wildly.*

Thanks for the good suggestion again. We have included an additional sentence.

6. *I found the text in the Figure 5 caption to be quite unclear. The caption states it is a plot between "a priori and a posteriori CHBr3 and CH2Br2 fluxes". I would recommend the authors to simplify the description to state it is a plot of X versus Y, i.e., in this case CH2Br2 fluxes versus CHBr3 fluxes. Then state the additional complexity that this plot includes both a priori (from two sources) and a posteriori data. As it is, too many things have been stated in a single sentence.*

We change the caption and the plot to make it clearer.

***Technical Comments***

1. *Page 3, line 89. Please change "The bottom-up approach assumes local flux estimates are representation of larger spatial scales." To "The bottom-up approach assumes local flux estimates are representative of larger spatial scales."*

Changed

2. *Page 6, line 194. Missing full stop after "respectively".*

Changed

3. Page 7, line 244. Missing comma after "e.g."

Changed

4. *Page 8, line 282. Remove "the" before "a posteriori".*

Changed

10 5. *Page 8, lines 280-284. Recommend changing the current text from "are more consistent with the observations that a priori fluxes." to "are more consistent with the observations than those from the a priori fluxes."*

Changed

6. *Page 10, lines 334. Recommend changing "particularly over north of tropics" to "particularly over the north of tropics"*

Changed

20 7. *Page 10, line 336. Missing "the" before "a priori"*

Changed

8. *Page 10, line 339. "coincide with large boundary layer measurements from CAST and CONTRAST.". Measurements of*
25 *what? It's perhaps better to be precise and state CH2Br2 given that the sentence starts by mentioning CHBr3.*

Changed

9. *Page 11, line 381. Change "use" to "uses".*

Changed

10. *Page 11, line 382. Change "develop" to "develops".*

Changed

*11. Figure 1. I would recommend that you label the individual panels with instructive headings such as CONTRAST and CAST.*

Titles have been added.

*12. Figure 2 caption. There is an unnecessary full stop in the second to last line of the caption.*

10 Changed

*13.  Figure 3. The panels are labelled a, b, c, etc. but these labels are not used in the caption. I recommend that you use the labels to improve clarity.*

Changed

*14. Figure 6. I find the caption description to be unclear. To simplify this I would recommend re-designing the plot by shifting the three plots in the column on the right hand side to be a new row underneath the a posteriori row. Call this new row a posteriori zoom, for instance. I would recommend moving the "Posterior" label to above the bottom row in Figure 6 rather below it. In addition, The Ordonez label in Figure 6 is barely readable. I would recommend moving the labels for*
20 *each column to be placed above the panels in black type font.*

Changed.

**Review report 2**

Many thanks for the kind supports and very good suggestions.  Our corrections are detailed below.

*Question:*

*Why is the agreement between model and observations better for CHBr3, which has a shorter lifetime, than for CH2Br2 (lines 24–26).*

As shown in Figure 2, due to the longer life time, model $CH_2Br_2$ concentrations are more prone to uncertainties in emissions outside our study domain, and to the errors in model long-distance transport.  Also, $CH_2Br_2$ is comparatively less temporal and spatial (vertical) variable than $CHBr_3$. As a result, the correlation between the model and observations is more sensitive to any model or measurement errors.

*Suggestions:*

*1. The addition of the new material illustrated in Figure 6 was very interesting and provides a very important finding (lines 400--401) that should be mentioned in the abstract and in the conclusion, namely that the choice of a priori plays a small role in determining a posteriori fluxes when atmospheric measurements are combined with the framework presented in this study (Geos-Chem an MAP).*

Thanks for the good suggestion. We have added  that in the abstract :

*2. Line 130: as with many acronyms properly defined throughout the manuscript, "GC/MS"*
*Should be spelled out as well.*

We now clarify it as:

'…the gas chromatography-mass spectrometry  instruments

*3. Lines 134--145: the two data sets (CAST and CONTRAST) are calibrated using different scales, as the authors described. Are there any differences in scales? If so, were*

*They accounted for in this analysis? The authors mentioned an offset between the two data sets (line 360-362) that were found when comparing to model*

*outputs. Could that be due to the difference in scales?*

Calibration of instruments operating within CAST and CONTRAST is the subject of Andrews et al, (2016), and we refer the reviewer to that dedicated paper. We use that adjusted CAST/CONTRAIL that show differences of about 7.7% at vertical range between 0-8km.  No CAST data are available above 8 km, so we cannot assess inter-instrument bias at higher altitudes.

*4. Lines 145--148: how were these errors determined? This work or Andrews et al? If the latter, then it should be referenced.*

They are determined by Andrew et al (2016), and the reference added as suggested.

*5. Line 351: "This model bias…". Which bias, the negative one? Clarify this statement.*

Changed to 'the small model negative bias…'  for clarification

*6. Multi-panel figures: consider using labels (e.g., (a), (b), (c)) in the panels and then add them to the captions for more efficient referencing. In some cases, for instance, Fig 3 and Fig 4, the labels were present in the panels, but not added to the captions.*

We have changed Figures 2 and 6, and the captions for Figure 3 and 4 to make them clearer for the reader.

*7. Figure 2: provide acronym in caption For "GC" Legend in rightmost panel.*

Changed.

*8. Figure 2: was the GC analysis for CHBr3 also done for CH2Br2? If so, what were the results?*

The paper is mostly focused on bromoform so we did not perform nested simulations for $CH_2Br_2$, as we find that CAST/CONTRAIL data provide better constraints on its local source estimates (Figure 3).

*9. Figure 2: the left panel shows that the model has a positive bias at all altitudes. The middle panel, however, shows both positive and negative biases, granted they are smaller in magnitude. What are the uncertainties Or error bars in these vertical profiles, which can help assess the relevance of the model-observation differences?*

Uncertainty in CAST/CONTRAST data usually varies with altitudes (Andrew et al., 2016). For simplicity, we have assumed a fixed uncertainty of 20% for $CHBr_3$ (taking into account model errors etc) in our inversion experiments. But observations at different altitudes have different sensitivities to local emissions, and hence it is possible that our a posteriori model concentrations show both positive and negative biases at different altitudes, in particular in the presence of (small) deviation between model and observation vertical lapse rates.

*10. Figure 6: are similar results obtained when examining CH2Br2? A few sentences on those results could be added to the text without the need to create new panels or figures.*

No similar experiment has been done for $CH_2Br_2$. Our current study is more focused on bromoform, as the CAST/CONTRAIL data provide better constraints on its local source estimates.

*11. Figure 6: include axes labels (lats/lons) in plots in the right.*

Changed.

[revised manuscript text omitted]